# From Genetic Mutations to Molecular Basis of Heart Failure Treatment: An Overview of the Mechanism and Implication of the Novel Modulators for Cardiac Myosin

**DOI:** 10.3390/ijms22126617

**Published:** 2021-06-21

**Authors:** Yu-Jen Chen, Chian-Shiu Chien, Chern-En Chiang, Chen-Huan Chen, Hao-Min Cheng

**Affiliations:** 1Department of Internal Medicine, Division of Cardiovascular Medicine, Wan Fang Hospital, Taipei Medical University, Taipei 116081, Taiwan; 100312@w.tmu.edu.tw; 2Department of Internal Medicine, Division of Cardiology, School of Medicine, College of Medicine, Taipei Medical University, Taipei 110301, Taiwan; 3Institute of Public Health, National Yang Ming Chiao Tung University, Taipei 112304, Taiwan; 4Innovative Cellular Therapy Center, Department of Medical Research, Taipei Veterans General Hospital, Taipei 112201, Taiwan; cschien6688@gmail.com; 5General Clinical Research Center, Taipei Veterans General Hospital, Taipei 112201, Taiwan; cechiang@vghtpe.gov.tw; 6Department of Medicine, National Yang Ming Chiao Tung University College of Medicine, Taipei 112304, Taiwan; 7Department of Medical Education, Taipei Veterans General Hospital, Taipei 112201, Taiwan; chench@vghtpe.gov.tw; 8College of Medicine, National Yang Ming Chiao Tung University, Taipei 112201, Taiwan; 9Center for Evidence-Based Medicine, Taipei Veterans General Hospital, Taipei 112201, Taiwan

**Keywords:** heart failure, dilated cardiomyopathy, hypertrophic cardiomyopathy, hereditary cardiomyopathy, cardiac myosin

## Abstract

Heart failure (HF) is a syndrome encompassing several important etiologies that lead to the imbalance between oxygen demand and supply. Despite the usage of guideline-directed medical therapy for HF has shown better outcomes, novel therapeutic strategies are desirable, especially for patients with preserved or mildly reduced left ventricular ejection fraction. In this regard, understanding the molecular basis for cardiomyopathies is expected to fill in the knowledge gap and generate new therapies to improve prognosis for HF. This review discusses an evolutionary mechanism designed to regulate cardiac contraction and relaxation through the most often genetically determined cardiomyopathies associated with HF. In addition, both the myosin inhibitor and myosin activator are promising new treatments for cardiomyopathies. A comprehensive review from genetic mutations to the molecular basis of direct sarcomere modulators will help shed light on future studies for a better characterization of HF etiologies and potential therapeutic targets.

## 1. Introduction

Heart failure (HF) is a complex syndrome characterized by substantial morbidity and mortality. HF affects millions of people worldwide, with an estimated prevalence of 1–2% of the adult population [1]. The global survival rates of HF remain poor, and the majority of patients with HF admitted to the hospital die within 5 years of admission. Although the introduction of modern evidence-based therapies and patient management systems have had progressive effects on outcomes, the hospital mortality rate remains approximately 2–17% for individuals with HF [2]. HF has been subclassified into three categories according to the left ventricular ejection fraction (LVEF): HF with preserved ejection fraction (HFpEF; in which the LVEF is ≥50%); HF with reduced ejection fraction (HFrEF; in which the LVEF is 40%); HF with mildly recued ejection fraction (HFmrEF; in which the LVEF is 41–49%) [3]. HFmrEF as heart failure with mildly reduced EF, rather than mid-range EF, may be a more appropriate term given the fact that the Prospective Comparison of angiotensin receptor-neprilysin inhibitor (ARNI) with angiotensin-receptor blockers Global Outcomes in HFpEF trial indicated that patients with LVEF lower than normal can still probably benefit from such therapies [4]. 

The most well-recognized compensatory mechanisms are activation of the sympathetic nervous system and renin-angiotensin-aldosterone system (RAAS). Neurohormonal activation initially compensates for impaired ventricular function to maintain hemodynamic stability when cardiac output is reduced. RAAS activation occurs with increased retention of salt and water, peripheral arterial vasoconstriction, and increased contractility of the left ventricle [5]. Despite the initial benefits of these mechanisms, prolonged activation contributes to deleterious effects on cardiac structure and symptoms of HF [6,7]. Standard medical therapy includes evidence-based beta-blockers (bisoprolol, carvedilol, and metoprolol succinate) and RAAS blockades, such as angiotensin-converting enzyme inhibitors (ACE-Is), angiotensin II receptor blockers (ARBs), and mineralocorticoid receptor antagonists (MRAs) [8,9]. Collecting evidence of HF with a particular focus on the study of its molecular basis is expected to lay the foundation for a better understanding of novel therapeutic agents. By contributing to a neutral endopeptidase, neprilysin, and its endogenous vasoactive peptides, the ARNI successfully reduced the risks of death and of hospitalization for HF with superiority to ACE-I [10]. The discovery of pacemaker current arouses the development of the selective sinus-node inhibitor, ivabradine, and confirms the importance of heart rate in the pathophysiology of HF [11]. Recently, inhibitors of sodium-glucose cotransporter 2 (SGLT2) reduced the risk of worsening HF or death from cardiovascular causes, possibly through glucose-independent mechanisms, regardless of the presence or absence of type 2 diabetes mellitus [12]. In addition, vericiguat, a novel oral soluble guanylate cyclase stimulator, decreased the incidence of death from cardiovascular causes or hospitalization among patients with high-risk HF [13]. We have summarized the evidence of available pharmacological therapy of HF in Table 1.

Although combined treatment with an ARNI, a beta-blocker, an MRA, and an SGLT2 inhibitor brings a significant improvement in patients with HFrEF (Figure 1), achieving functional and structural recovery is distant. In addition, the lack of effective treatments for HFpEF represents a large, unmet need in cardiology. Potential mechanisms remain to be elucidated, such as the use of SGLT-2 inhibitors in HF populations. Both hypertrophic cardiomyopathy (HCM) and dilated cardiomyopathy (DCM) are involved in cardiomyocyte contractility and relaxation, and their regulation, regardless of the etiology, is the main cause of the progression of symptomatic HF [14]. Hence, a comprehensive review of cardiomyocyte abnormalities in a failing heart helps tailor-specific therapies to improve prognosis. This text reviewed the gene mutations of HF on the molecular basis, with a focus on cardiomyocytes.

**Table 1 ijms-22-06617-t001:** Evidence of available pharmacological therapy of heart failure.

Medications	Initial Dose	Target Dose	Trial	Year
**ACE-I**				
Enalapril [15]	2.5 mg BID	20 mg BID	CONSENSUS	1987
Fosinopril [16]	5–10 mg daily	40 mg daily	FEST	1995
Captopril [17]	6.25–25 mg TID	50 mg TID	SAVE	1992
Enalapril [18]	2.5 mg BID	20 mg BID	SOLVD	1991
Perindopril [19]	2 mg daily	16 mg daily	PEP-CHF	2006
Ramipril [20]	1.25–2.5 mg daily	10 mg daily	AIRE	1993
Trandolapril [21]	1 mg daily	4 mg daily	TRACE	1995
**ARB**				
Candesartan [22]	4–8 mg daily	32 mg daily	CHARM	2004
Losartan [23]	25–50 mg daily	150 mg daily	ELITE II	2000
Valsartan [24]	20–40 mg BID	160 mg BID	Val-HeFT	2001
**Beta Blockers**				
Bisoprolol [25]	1.25 mg daily	10 mg daily	CIBIS-II	1999
Carvedilol [26]	3.125 mg BID	50 mg BID	Carvedilol Heart Failure Study	1996
Metoprolol [27]	12.5–25 mg daily	200 mg daily	MERIT-HF	1999
Nebivolol [28]	1.25 mg daily	10 mg daily	SENIORS	2005
**MRA**				
Spironolactone [29]	12.5 mg daily	25 mg daily	RALES	1999
Eplerenone [30]	25 mg daily	50 mg daily	EMPHASIS-HF	2010
**Vasodilators**				
Hydralazine +Isosorbide dinitrate [31]	25 mg TID20 mg TID	300 mg daily160 mg daily	V-HeFT	1986
**I_f_ Inhibitor**				
Ivabradine [11]	5 mg BID	7.5 mg BID	SHIFT	2010
**ARNI**				
Sacubitril/valsartan [10]	49 mg/51 mg BID	97 mg/103 mg BID	PARADIGM-HF	2014
**SGLT-2 inhibitors**				
Dapagliflozin [12]	5 mg daily	10 mg daily	DAPA-HF	2019
Empagliflozin [32]	10 mg daily	10 mg daily	EMPEROR-Reduced	2020
**Guanylate cyclase stimulator**				
Vericiguat [13]	2.5 mg daily	10 mg daily	VICTORIA	2020

ACE-I, angiotensin-converting enzyme inhibitor; ARB, angiotensin-II receptor blocker; MRA, mineralocorticoid receptor antagonist; ARNI, angiotensin receptor–neprilysin inhibitor; SGLT-2, sodium–glucose cotransporter 2; CONSENSUS, Cooperative North Scandinavian Enalapril Survival Study; FEST, Fosinopril Efficacy/Safety Trial; SAVE, Survival and Ventricular Enlargement; SOLVD, Studies of Left Ventricular Dysfunction; PEP-CHF, Perindopril in elderly people with chronic heart failure; AIRE, Acute Infarction Ramipril Efficacy; TRACE, Trandolapril Cardiac Evaluation; CHARM, Candesartan in heart failure assessment of reduction in mortality and morbidity; ELITE II, Losartan Heart Failure Survival Study; Val-HeFT, The Valsartan Heart Failure Trial; CIBIS-II, The Cardiac Insufficiency Bisoprolol Study II; MERIT-HF, Metoprolol CR/XL Randomised Intervention Trial in Congestive Heart Failure; SENIORS, Study of the Effects of Nebivolol Intervention on Outcomes and Rehospitalisation in Seniors with Heart Failure; RALES, Randomized Aldactone Evaluation Study; EMPHASIS-HF, Eplerenone in Mild Patients Hospitalization and Survival Study in Heart Failure; V-HeFT, Vasodilator Heart Failure Trial; SHIFT, Systolic Heart Failure Treatment with the If inhibitor ivabradine trial; PARADIGM-HF, Prospective Comparison of ARNI with ACEI to Determine Impact on Global Mortality and Morbidity in Heart Failure; DAPA-HF, Dapagliflozin and Prevention of Adverse-Outcomes in Heart Failure; EMPEROR-Reduced, Empagliflozin Outcome Trial in Patients With Chronic Heart Failure With Reduced Ejection Fraction; VICTORIA, Vericiguat Global Study in Subjects with Heart Failure with Reduced Ejection Fraction.

## 2. Cardiomyocyte Abnormalities in HF

The heart is composed of the ventricular cardiomyocytes, atrial cardiomyocytes, pacemaker cells, Purkinje cells, blood vessels, and connective tissue. Billions of ventricular cardiomyocytes make up the left ventricle, accounting for approximately 50% of the heart mass. Myofibrils are comprised of actin thin filaments, myosin thick filaments, and titin, which anchor the myosin thick filament to the Z-disc in the sarcomere, forming contractile elements of cardiac muscle. Transverse tubules (T-tubules) extend from the surface throughout the cells and permit rapid transmission of the action potential. When the cardiac action potential propagated from sinoatrial node pacemaker cells reaches the T-tubules, membrane depolarization opens voltage-gated L-type Ca^2+^ channels. The initial influx of Ca^2+^ further stimulates the efflux of Ca^2+^ from ryanodine receptor type-2 (RyR2) channels on the sarcoplasmic reticulum, resulting in contraction of the heart (Figure 2) [33]. In addition to myofibril, the contractile apparatus contains tropomyosin and the troponin complex, which is composed of three subunits: troponin C, I, and T. These proteins regulate cross-bridge formation depending on the Ca^2+^ concentration [34]. An increased Ca^2+^ binding to troponin C leads to a conformational change in the troponin complex and promotes actin-myosin cross-bridge cycle in the systole. Troponin I and T block myosin-binding sites on actin in decreased Ca^2+^ of the diastole. Ca^2+^ reuptake by sarcoplasmic/endoplasmic reticulum Ca^2+^-ATPase 2a (SERCA2a) stimulates active relaxation of cardiomyocytes, and the expression of SERCA2a are significantly reduced in failing hearts [35].

Cardiomyopathy is a disease of functional abnormality of the cardiac muscle with progressive cardiomyocyte loss, and eventually leads to HF by means of necrotic, apoptotic, or autophagic cell death pathways. HCM and DCM are the two major types of intrinsic cardiomyopathies and are the most often genetically determined cardiomyopathies associated with HF [36]. Approximately 40–60% of cases of HCM and DCM have been recognized as genetic diseases with mutations in sarcomere [37,38] that are involved in cardiomyocyte contractility and its regulation. Both HCM and DCM decrease the stroke volume. HCM is characterized by pathological left ventricular hypertrophy, accompanied by myofibril disarrays and a decrease in myofibril density, leading to a decrease in force in isolated cardiomyocytes [39]. Hence, the ejection fraction in patients with HCM increases through reduced ventricular cavity volume, rather than increased contractility. In contrast, DCM is characterized by a dilated and thin ventricular wall with systolic impairment. However, the force production from cardiomyocytes was recognized only in one study with a mutation in lamin A/C [40]. Therefore, the ventricular cavity volume is the major determinant causing the ejection fraction to be significantly lower in DCM than in HCM. Although the etiology of primary cardiomyopathy remains unclear, various genetic mutations associated with primary cardiomyopathy have been uncovered [41]. In addition, factors such as proteasome dysfunction [42], sarcomeric protein isoform transitions [43], and altered protein phosphorylation [44] are related to primary cardiomyopathy.

## 3. Molecular Basis of Hereditary Cardiomyopathy

Although the overall cardiac force of contraction is modulated by multiple factors, changes in sarcomere length is an important mechanism, referred to as length-dependent activation [45]. With increased sarcomere length, there is an increase in the force of contraction and an augmentation in the number of myosin molecules available for cross-bridge formation, causing adenosine triphosphate (ATP) consumption. Myosin contains two heads with myosin ATPase activity, which is an enzyme that hydrolyzes ATP required for cross-bridge formation. Dynamic conformational changes between paired myosin molecules predict contractility effects [46,47]. During relaxation, head domains of paired myosin molecules construct interacting-heads motif (IHM) [48] that promote myosin in an energy-conserving super-relaxed state (SRX) conformation, which has extremely slow ATP hydrolysis and thereby changes the number of myosin heads that interact with actin or a disordered relaxed state (DRX) conformation, in which only one myosin head is available to hydrolyze ATP and interact with actin [49,50].

Approximately 50% of cases of HCM are identified as genetic disease-causing mutations. ß-myosin heavy chain 7 (MYH7) and myosin-binding protein C (MYBPC3) are the most frequent sarcomere mutations, accounting for approximately 80% of all HCM cases [51]. Although other sarcomeric genes are less frequently implicated, mutations in any components of sarcomere, such as α-tropomyosin gene, cardiac troponin T gene (TNNT2), ventricular myosin essential light chain gene, ventricular myosin regulatory light chain 2 (MYL2) gene, cardiac troponin I gene, cardiac α-actin, cardiac troponin C, and cysteine and glycine-rich protein 3, can result in HCM [41,52,53,54,55,56,57,58]. The analysis of functional changes related to mutations in sarcomere components, including myosin heavy chains, α-tropomyosin, and cardiac troponin T, mainly focuses on decreased power generation [59,60]. The decreased power was not the core of functional change caused by the sarcomere mutations. Instead, most sarcomere mutations in HCM exhibited increased Ca^2+^ sensitivity of force development without increasing maximal active force levels [61,62,63]. The concentration of intracellular Ca^2+^ plays a central role in regulating cross-bridge formation. Intracellular Ca^2+^ released from the sarcoplasmic reticulum RyR2 promotes the actin-myosin interaction in the systole, and reuptake via SERCA2a stimulates relaxation during the diastolic phase. Myosin undergoes physiological shifts between the SRX conformation that maximizes energy conservation and the DRX conformation that enables cross-bridge formation with a greater ATP consumption [46]. These pathogenic myosin missense mutations evoke hypercontractility and impaired relaxation, causing hypertrophic remodeling with increased energetic stress by the heart, and eventually lead to HF. Studies have demonstrated that pathogenic MYH7 missense variants may destabilize the IHM and disrupt the physiological balance of myosin in the SRX and DRX conformations, which were associated with higher rates of HF in patients with HCM [46,64,65].

Unlike relatively frequent mutations in Z-disc components, cardiac myosin missense mutations account for approximately 10% of cases of primary DCM [66]. Both mutations of the cardiac myosin heavy chain and MYL2 are involved in primary DCM [67,68]. Mutant cardiac myosin heavy chain genes exhibit a reduced ability to translocate actin and a reduced actin-activated ATPase activity that causes remodeling and pathological dilation of the heart [69]. Mutations in the MYL2 gene may result in aberrant cardiomyocyte cytoarchitecture, leading to ventricular chamber dilation and decreased contractility [70]. Sarcomere mutations can be identified in both HCM and DCM with a certain degree of genetic overlap, but with differences in the functional phenotypes [71]. On a molecular basis, there are differences in protein folding and fiber assembly between actin mutations in HCM and DCM [72]. Another study demonstrated that TNNT2 mutations were found in both HCM and DCM with two distinct mechanisms that change the Ca^2+^ sensitivity of cardiac muscle contraction in opposite directions [73].

## 4. Small-Molecule Modulator of Cardiac Myosin

Mavacamten is a first-in-class, small molecule, selective allosteric modulator of cardiac myosin ATPase that targets the underlying sarcomere hypercontractility of HCM by reducing actin-myosin cross-bridge formation [74,75]. Although there is a close correlation between the rate of force generation in the cross-bridge and ATPase activity of β-cardiac myosin, inhibition of the increased ATPase activity of β-cardiac myosin reduces the overall activity, and thus alleviates the pathological phenotype. Green et al. reported that mavacamten reduced the ATPase activity in a dose-dependent manner and acted directly against cardiac myosin [76]. Furthermore, treatment with mavacamten inhibited phosphate release from β-cardiac myosin-S1 without slowing adenosine diphosphate (ADP) release, and thereby decreased the ensemble force and contractility produced by the sarcomere [76]. Mavacamten also inhibited steady-state actin-associated and actin-independent ATPase cycling in permeabilized cardiac myofibrils. It also attenuated hypertrophic and profibrotic gene expression in mouse models of HCM [76]. These findings provided evidence for the mechanism of mavacamten: inhibition of actin-activated phosphate release, the kinetic step most tightly coupled to force generation, and a direct effect on the binding of myosin-S1 to actin in the transition state [74].

SRX is observed in myosin filaments and it plays a central role in modulating force generation and energy use within the sarcomere. Modulation of myosin function through SRX is a flourishing theory in the therapeutic development of HF. Modulation of myosin SRX brings mavacamten to the clinic [77]. While myosin filaments are in close proximity to actin filaments and are able to hydrolyze ATP, they drive the sliding of thick and thin filaments and contracting sarcomeres that allow the heart to circulate blood. SRX is a biochemical state in which myosin heads are away from the thin filaments, providing a more ordered structure with a low metabolic rate in relaxed cells [46]. The properties of the SRX differ dramatically between cardiac and skeletal muscles. Hooijman et al. demonstrated a rapid transition of myosin heads out of the SRX in active skeletal fibers, whereas the population of the SRX remained constant in active cardiac cells [64]. In this manner, the cardiac muscle maintains a steady contraction and relaxation on an average rate over the lifetime in a graded and tightly moderated way. A certain proportion of myosin heads remain in the SRX conformation, while the missense variants in MYBPC3 reduce SRX proportions in the cardiac muscle and are associated with HCM [78]. A recent study showed that mavacamten stabilizes an autoinhibited state of two-headed cardiac myosin in tissue-purified solution [79]. This finding provided a new aspect of the mechanism that was not previously observed in studies of single-headed myosin. Mavacamten increases myosin SRX conformations [65,74,80]. These evidences indicated that mavacamten may decrease the metabolic load on the heart by being cardioprotective, particularly in a time of stress through the SRX (Figure 3A).

## 5. Novel Selective Cardiac Myosin Activator

HFrEF accounts for approximately 50% of HF cases worldwide. The pathogenesis of HFrEF is dysfunction of cardiomyocytes and, consequently, a decrease in cardiac contractility [81]. The initial reduction in cardiac systolic function increases cardiac wall stress, which results in neurohormonal activation and cardiac remodeling.

The compensatory mechanisms increase myocardial oxygen consumption, which causes myocardial injury, cardiomyocyte death, and eventually symptomatic HF. Improving systolic function may reverse or attenuate the negative feedback mechanism of HF. However, current guide-directed medical therapies improve the survival of patients with HF alone without demonstrating improvement in cardiac systolic function and ventricular volume. Although inotropic agents, including beta-adrenergic agonists, dopamine and alpha-adrenergic receptor agonists, and phosphodiesterase-3 inhibitors, have shown the ability to increase cardiac contractility by increasing calcium influx or reducing the degradation of cAMP, none of these agents are able to improve the outcomes of HF, and are supposed to increase mortality [82,83,84,85,86,87,88].

Omecamtiv mecarbil (OM), a novel selective cardiac myosin activator, increases the inotropic support without the adverse effects of conventional inotropic agents [89]. Physiologically, the ATP is hydrolyzed from ATP to ADP and inorganic phosphate (Pi) by the ATPase. The free myosin head moves into position on actin binding sites and forms the “pre-powerstroke” state, which is ready for actin-myosin cross-bridge formation. While the Pi releases from myosin, the remaining myosin-ADP complex firmly attaches to actin, triggering the “powerstroke,” which is another conformational change that pulls myosin against actin to generate forces, shortening the sarcomere (Figure 3B). OM increases the rate of Pi release and binds to an allosteric site that stabilizes the pre-powerstroke state of myosin, enabling more cardiac myosin heads to undergo a powerstroke prior to the onset of systole [90]. Besides, OM has a potentially favorable energetic efficiency by reducing ATP consumption in the absence of an interaction with the actin filament, and directly acts on the myofilament (myotropes) without altering the cardiomyocyte calcium transient (calcitropes) [91,92]. Given the lack of change in intracellular calcium concentrations, OM improves ventricular systolic function without the adverse effects of conventional inotropic agents, which are mainly caused by the mechanism of calcitropes.

The Acute Treatment with Omecamtiv Mecarbil to Increase Contractility in Acute HF study (ATOMIC-AHF) firstly demonstrated that intravenous OM increased the systolic ejection time, systolic blood pressure, and decreased the heart rate and left ventricular end-systolic dimensions [93]. Further, the Chronic Oral Study of Myosin Activation to Increase Contractility in HF (COSMIC-HF) showed similar effects of oral OM [94]. Other studies have demonstrated that the LV systolic time is decreased by 10 to 70 ms in cardiac dysfunction, and is an independent predictor of incident HF and the risk of mortality [95,96]. OM shifts HF patients toward a normal ejection duration and increased stroke volume, as well as decreases the end-diastolic dimensions. OM may also reduce ventricular wall stress, as reflected in the decreased N-terminal pro–B-type natriuretic peptide [97], and sympathetic withdrawal, supported by the decreased heart rate [98].

## 6. Conclusions

HF remains a crucial public health problem worldwide, and is an ongoing drain on healthcare expenditures. Although a variety of medications and interventional procedures are available, innovative strategies addressing the core of the disease, rather than ameliorating the symptoms, are essential. In this review, we summarized the gene mutations to the molecular basis of HF with a focus on cardiomyocytes to shed light on further precise cardiac delivery of either novel or clinically available drugs together. To achieve a more complete understanding of HF, studying its basic science provides an invaluable resource with which to achieve better diagnoses, new aspects of pathophysiology, and novel therapeutic targets of HF.

## Figures and Tables

**Figure 1 ijms-22-06617-f001:**
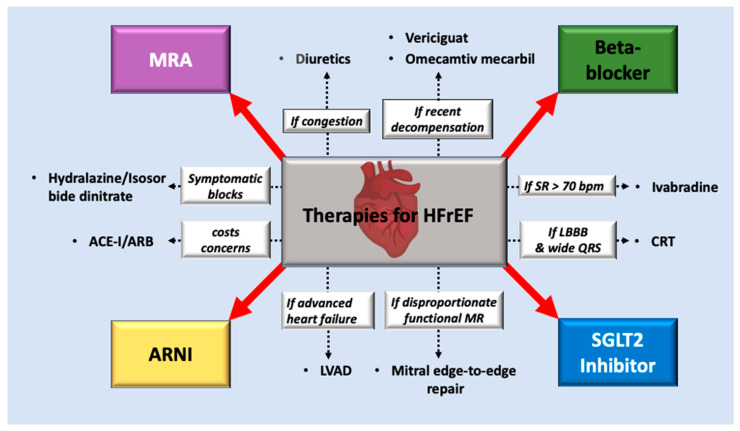
Pharmacological and interventional treatment for heart failure with reduced ejection fraction: The red arrows indicate a combination of the four drugs that should be initiated early in HFrEF patients. The dotted arrows indicate the additional drugs and interventional therapies for individualized treatment for specific populations.

**Figure 2 ijms-22-06617-f002:**
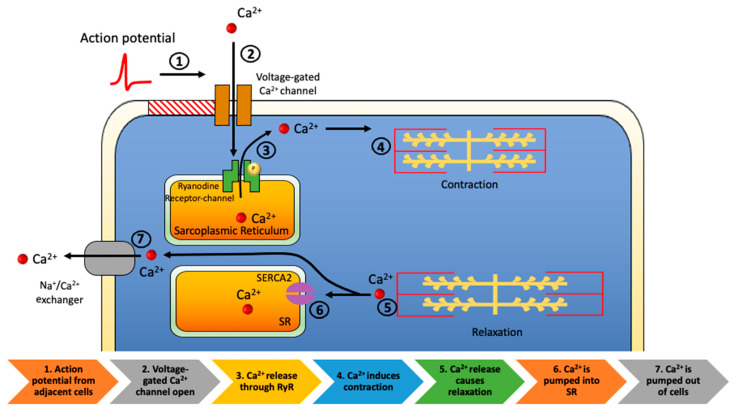
Cardiac excitation-contraction coupling: Action potential transmits directly from adjacent cells (step 1). Calcium enters cardiomyocyte through voltage-gated L-type Ca^2+^ channels (step 2). The initial influx of Ca^2+^ stimulates efflux of Ca^2+^ from ryanodine receptor type-2 channels on the sarcoplasmic reticulum (step 3) resulting in contraction of the heart (step 4). As the contraction ends (step 5), the Ca^2+^ reuptake is completed by sarcoplasmic/endoplasmic reticulum Ca^2+^-ATPase 2a (SERCA2a) (step 6). The sodium-calcium exchanger removes Ca^2+^ from cells (step 7).

**Figure 3 ijms-22-06617-f003:**
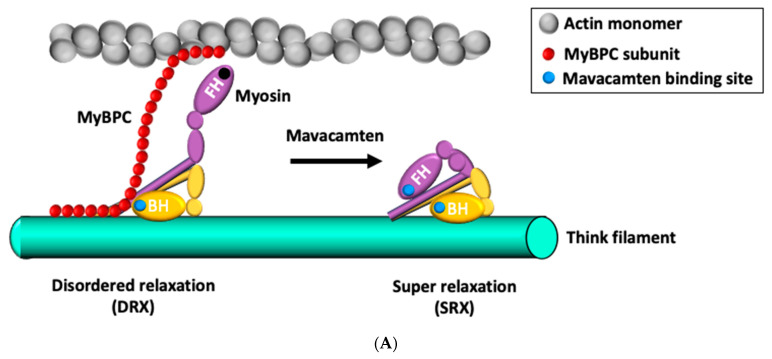
(Panel **A**): A schematic of two myosin conformations and associated energy consumption throughout the cardiac cycle: A pair of myosin molecules is depicted, with each head denoted as blocked (BH) or free (FH). The disordered relaxed state (DRX) insulates the BH, resulting in inhibition of ATPase (denoted by a blue dot), while FH retains adenosine triphosphate (ATP) hydrolysis activity and triggers actomyosin function modulating by myosin binding protein C (MyBPC). The super-relaxed state (SRX) insulates both BH and FH, and energy conservation is maximized because both myosin ATPase domains are inhibited. Hypertrophic cardiomyopathy variants preferentially shift conformation toward the DRX, resulting in hypercontractility, impaired relaxation, and excessive energy consumption. Mavacamten can stabilize the SRX state, attenuate hypercontractility, and improve compliance in cardiomyocytes. (Panel **B**): Biochemical cycle of cardiac myosin. During diastole, one head of myosin hydrolyzes ATP to adenosine diphosphate (ADP) and inorganic phosphate (Pi) (Step 1). Actin becomes accessible, and the head of myosin binds to it and forms the “pre-powerstroke” state. During systole, a subset of myosin heads in the pre-powerstroke promotes the release of Pi (Step 2), which initiates the “powerstroke” to generate force, shortening the sarcomere. (Step 3). The loss of ADP and another binding of ATP (Step 4) releases the myosin from the actin filament because myosin-ATP has a low affinity for actin (Step 5). Omecamtiv mecarbil increases the rate of Pi release (Step 2) and bias the ATP hydrolysis step (Step 1) toward pre-powerstroke state, enabling more myosin heads to undergo a powerstroke during systole.

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
