# Peer review of "From Genetic Mutations to Molecular Basis of Heart Failure Treatment: An Overview of the Mechanism and Implication of the Novel Modulators for Cardiac Myosin"

_ijms, 2021, doi:10.3390/ijms22126617_

Round 1

Reviewer 1 Report

Please show us the more specific and understandable figure (Fig1~3).

Reviewer 2 Report

Thank you for permitting me to review this manuscript 

Abstract 

Please insert a couple of sentence on new medications 

Introduction 

Line 42-45 please provide reference 

LIne 42-48. PLease rephrase by making 2  phrases the current one is too long

Line 49-53. please provide reference 

evidence based betablocker please explain 

Figure 1  must be improved by the increase in the size of the police currently it is very difficult to read 

Figure 2 Please locate ryanodine receptor and insert numbers or sequence for respective events 

A more  concrete illustration would be helpful to better understand the new molecules such as mavacamtem right now  this is not the case 

Line 276  Please provide reference

conclusion 

I don't see much reference in the text on gene mutations  the authors  speaks mainly about molecular mechanisms please insert some hints to gene mutations otherwise delete the sentence with reference to gene mutations in the title or in the conclusion 
